# Measuring Quality of Life in Parkinson’s Disease—A Call to Rethink Conceptualizations and Assessments

**DOI:** 10.3390/jpm12050804

**Published:** 2022-05-16

**Authors:** Maria Stührenberg, Carolin S. Berghäuser, Marlena van Munster, Anna J. Pedrosa Carrasco, David J. Pedrosa

**Affiliations:** 1Department of Neurology, University Hospital Marburg, 35043 Marburg, Germany; stuehren@students.uni-marburg.de (M.S.); berghaeu@students.uni-marburg.de (C.S.B.); marlena.vanmunster@uni-marburg.de (M.v.M.); 2Research Group Medical Ethics, Philipps-University Marburg, 35043 Marburg, Germany; anna.pedrosacarrasco@uni-marburg.de; 3Center of Mind, Brain and Behaviour, Philipps-University Marburg, 35041 Marburg, Germany

**Keywords:** Parkinson’s disease, quality of life, measurement, assessment, opinion

## Abstract

Parkinson’s disease (PD) is a chronic condition that considerably impacts the perception of quality of life (QoL) in both patients and their caregivers. Modern therapeutic approaches and social efforts strive at maintaining and promoting QoL. It has emerged as a fundamental parameter for clinical follow-up and poses one of the most important endpoints in scientific and economic evaluations of new care models. It is therefore of utmost importance to grasp concepts of QoL in a meaningful way. However, when taking a look at the origin of our modern understanding of QoL and existing methods for its measurement in PD patients, some aspects seem to lack sufficient appreciation. This article elaborates on how the perception of health and QoL have changed over time and discuss whether current understandings of both are reflected in the most commonly applied assessment methods for people with PD.

## 1. Introduction

Parkinson’s disease (PD) is a chronic condition that places a high burden on both patients and their caregivers. Worldwide, approximately 7 to 10 million people suffer from PD, a prevalence that is likely to increase significantly in future societies according to demographic changes to come [1,2]. Given its incurability to date, modern therapeutic approaches and social efforts primarily strive at maintaining and promoting patient-related quality of life (QoL) [3]. It has evolved as a fundamental parameter for clinical follow-up and it poses one of the most important endpoints in scientific and economic evaluations of emerging care models [4,5,6,7]. It is therefore of utmost importance to grasp concepts of QoL in a meaningful way. However, when contemplating the origins of our current understanding of QoL and the existing methods to measure it in PD patients, some aspects may not be sufficiently appreciated. The existing literature has already critically examined established survey methods and the understanding of QoL in the field of Parkinson’s research [8], but what is missing so far is a perspective on how the assessment of QoL can be improved. In what follows, we elaborate on how the perception of health and QoL has changed over time and discuss whether current understandings of both are reflected in the most commonly applied assessments for people with PD. Finally, we would like to provide food for thought on how conventional QoL instruments could potentially be refined in the future to personalize applications and ultimately improve patient care.

## 2. Theorizing Quality of Life in Parkinson’s Disease

### 2.1. Conceptualizations of Patient-Related Quality of Life: A Reflection of the Health Paradigm

Effectiveness and safety constitute important domains of healthcare quality [9]. Nevertheless, over the last decades, what we consider “good” healthcare is undergoing a major transformation, leaving behind the mere contemplation of morbidity and mortality [10] and devoting more attention to the third pillar of healthcare quality, people-centeredness [9]. These changes have profound implications for chronic neurodegenerative diseases such as PD and may entail a fundamental paradigm shift for future care concepts.

A paradigm can be understood as a set of rules which guide beliefs, habits, and procedures in society [11]. In the public health and medical context, “health” represents one of the most eminent paradigms [11,12]. “Health” can be understood as a paradigm insofar as the idea of health dictates how topics are discussed, how certain things are thought about, and how decisions are made. For example, what is considered to be good health status determines decisions in clinical treatment processes [13]. On a society level, the paradigm of health is central to the understanding, the design of, and practice within healthcare services. Accordingly, our perception of “good” healthcare is firmly attached to our understanding of the purpose that this type of care serves, namely, health. Nowadays, it is widely recognized that individual health status is not solely determined by the absence of disease but by a combination of physical, “mental and social well-being” [14]. As a result of this paradigm evolution, the focus in healthcare began to shift from promoting quantity to fostering QoL in the 1960s [15]. From the 1980s onwards, interest in QoL as a decision-making criterion grew [15], culminating in a new debate from the 1990s onwards that framed these concepts as a subjectivist notion and eventually led to inclusion of concepts such as happiness [15]. This glance at history suggests that what exactly is understood by QoL is defined differently and changes over time.

Table 1 outlines key conceptualizations from three decades and shows how the recognition of an ethical construct grew into a multidimensional concept that aims at reflecting an individual’s subjective evaluation. This evaluation may be affected by individual factors, such as one’s educational background or the perceived quality of main activities in life (e.g., quantity and quality of employment, presence of leisure activities), but also by social aspects, such as the presence of basic human rights or the presence of social support [16]. Summarized, QoL may be understood as the difference between an individual’s hopes and expectations and their experience of life at a given point in time [17].

In addition to this rather broad concept of quality of life, health-related quality of life (HRQoL) is a component of QoL related to health and to care. Overall, HRQoL describes how individual QoL may be affected over time by a disease, disability, or disorder [10]. It was suggested that HRQoL encompasses the following dimensions:physical function;role function (e.g., work, chores);cognitive function;psychological function;social function (e.g., relationship with caregivers, intimacy);spiritual well-being;physical symptom burden.

The relative importance of each dimension may vary intra- and interpersonally as time passes [21]. Other definitions of HRQoL focus on patients’ social, emotional, and physical well-being following treatment or characterize the impact of a person’s health on their ability to live a fulfilling life [22]. This all together emphasizes how transient concepts of QoL are and therefore the importance of an ample insight into different aspects of life, especially when neurodegenerative diseases successively deprive people of their physical resources.

Although the fundamental work on QoL lies beyond the scope of PD research, the knowledge gained can in theory be easily transferred to this disease. In accordance with the subjectivist understanding of QoL, modern PD care aims not only to alleviate symptoms but also to adopt a holistic view of patients’ individual psychological health and living condition [23]. This is not least reflected in the process of healthcare evolution towards integrated care models that address patients’ needs holistically and take individual factors, such as living circumstances, into consideration [4,5,6,7]. Measurements of HRQoL and QoL are crucial to PD-related healthcare research, audit, and practice and are sought through numerous survey instruments [24]. However, it remains debatable as to whether these instruments have kept pace with the abovementioned evolution of QoL conceptualizations and the models of PD care by implementing subjective, individual aspects of life.

### 2.2. Measuring Quality of Life in Parkinson’s Care—Is There a Need for Improvement?

Measuring QoL in PD takes place in three principal areas of application: clinical management, clinical audit, and research. Assessment of QoL can be used as an adjunct to routine clinical assessment to foster individual patient care. Audits may enable assessments of whether processes, requirements, and policies of healthcare services meet required standards. Usually, as part of a quality management process, audits allow for capturing trends and for providing feedback on the effectiveness of interventions that have been initiated. In this context, (serial) QoL data may serve as a source of information for resource allocation. Lastly, given the increasing awareness of the importance of patient-reported outcomes, QoL is a frequently reported endpoint in clinical trials. The demands made on the instruments depend on their purpose. For clinical routine, short, concise questionnaires are needed in order to guarantee a rapid and effortless assessment, whereas for research purposes, it may be appropriate to use lengthier and more complex instruments in order to adequately address research questions. However, among all instruments, one questionnaire is used particularly frequently: the 39-item Parkinson’s Disease Questionnaire (cf. Table 2). Irrespective of the purpose it serves, many QoL measures for PD are tailored to quantify symptom burden, as well as physical and psychological function, thereby somehow neglecting the aforementioned fundamental aspects of QoL.

Coming back to the concepts of QoL and HRqoL, one may argue against questionnaires predominantly assessing HRqOL and not QoL. One rationale for this may be the fact that dimensions of HRqOL, such as symptom burden, are easier to define and to assess. Additionally, results of HRQoL measurements might be more valuable for clinical practitioners, as they concentrate on challenges related to the disease and treatment. However, why is it then that studies and questionnaires keep referring to QoL and not HRQoL? In the literature, the term QoL is not used unequivocally, which also affects the quality of measuring instruments. In a review by Den Oudsten and colleagues [8], eight different PD-specific and six general QoL measurements were checked for their accuracy in actually measuring QoL. Only one of the specific and one of the general tools examined QoL, while the remaining aimed at related concepts, such as health status [8].

This ambiguity between concepts is problematic as it implies that clinicians and researchers might use measurement tools under the misapprehension that they can make a statement about QoL, while in fact they only retrieve information about their patient’s HRQoL or about related concepts. Why then are we appealing for the need to measure QoL instead of HRQoL?

The assumption of PD patients with higher symptom burden suffer from stronger impacts on HRQoL is not wrong [26]. At the same time, the expectation of an inevitable effect on their overall QoL may be questionable. Purely focusing on HRQoL neglects the subjectivity and multidimensionality of life [26,27]. In simple clinical settings, considering HRQoL may be sufficient, but in complex care situations, the manifold aspects of health are disregarded. Hence, the quality and the effectiveness of modern integrated PD care that addresses the patient’s living conditions holistically may require different survey tools with the ability to portray this.

According to a (modern) subjectivist understanding of health, the individual assessment of a situation is an important determinant of QoL. The perception of changes in one’s own physical abilities strongly depends on individual resources and values. Given that many PD patients can adapt to their new reality and even report sort of “disease gains” despite significant disabilities corroborates this assumption [28]. Additional factors to take into account that may influence perceptions of symptoms and their consequences on life are psychological factors such as a tendency towards dispositional optimism [29,30,31,32] or resilience [33,34]. Resilience is “related to processes and skills that result in good individual and community health outcomes, in spite of negative events, serious threats and hazards” [35]. These notions are supported by the results of Robottom and colleagues who found that resilience was not associated with disease severity in PD [36]. Factors such as disease gain, optimism, and resilience might therefore fundamentally shape the subjective evaluation of one’s own QoL, and some attention should be paid to how they can be taken into account when measuring QoL.

## 3. Moving towards a Holistic Measurement of Quality of Life in Parkinson’s Disease: Where Do We Go from Here?

Having discussed the evolution of QoL, its definition over the course of time, and the differences between HRQoL and QoL, but also shortcomings of current measurement tools, the final section of this paper addresses some potential alternatives to estimate QoL of PD patients.

Each person’s perceived QoL is highly individual, so designing a comprehensive measurement tool is anything but trivial. As a starting point, clear definitions and strict separation from HRQoL is required. In terms of the difficulty of mapping qualitative characteristics, one may advocate for qualitative assessment methods, e.g., in the form of structured interviews with open questions as a possible remedy [37]. These techniques may be particularly helpful to define additional dimensions to those implied in HRQoL questionnaires. Defining such distinct dimensions may enable the identification of patient-reported themes that need mentioning and should be assessed by holistic PD-related QoL assessments. For example, qualitative studies involving people with chronic illness recognized self-acceptance of the disease as a significant impact factor on QoL perception [38,39]. Albrecht and Devlieger describe that patients who succeed in adapting to their “new” life re-evaluate old ideas and values, which enables them to perceive their life as meaningful [26]. On the basis of this finding, the authors identified a need to include items on the topics of “acceptance of unchanging circumstances”, “acceptance of oneself”, or “resistance to illness” as essential overarching topics in QoL questionnaires [26]. Unfortunately, these factors that are closely linked to resilience are frequently overlooked in QoL instruments in PD.

The benefit of structured interviews may not yet obscure the fact that this technique is hardly practicable in everyday clinical practice. This raises the question if there are also better ways to assess individual QoL by means of quantitative approaches.

Particularly in clinical settings but also for scientific evaluations, there is a need for concise and applicable measurements that may be both reliable and may provide relatively quick results. One possible solution may lie in the development of new instruments, taking into account additional aspects of quality of life, e.g., perceiving patients in their social network including hobbies or close relationships [40]. An intriguing approach to measure QoL more accurately with quantitative measures, which has already been successfully tested for the population of patients with amyotrophic lateral sclerosis [41], would be to devise a questionnaire that allows patients to decide which dimensions are most important to them. By self-weighting the dimensions, a more subjective assessment of QoL could be achieved, which would simultaneously respond to changing priorities over time.

Given the resource-intensive procedure of developing questionnaires, the last option at least to bridge the time in between could be optimizing established quantitative questionnaires (e.g., on the basis of qualitative findings) in such a way that they depict more accurately what influences the QoL of people with PD.

A natural consequence of the holistic assessment of quality of life, however, will also present clinicians with the challenge of doing justice to the individual domains measured. It equally obliges looking beyond a purely symptom-oriented medicine and addressing the manifold problems of PD patients and their relatives in a multi-professional team at best in order to improve patients’ quality of life.

## 4. Conclusions

In summary, rethinking the nature of health and healthcare quality also requires a re-evaluation of the tools used in clinical practice, audits, and research. QoL in this context reflects an integral aspect of patient-centeredness, which needs to be made measurable in an adequate way. In the coming years, a scientific focus must therefore be placed on better mapping QoL in questionnaires in order to meet the needs of patients with PD in the continuum of care.

## Figures and Tables

**Table 1 jpm-12-00804-t001:** Key conceptualizations of QoL over the course of the last decades.

Source	Definition	Comments
Elkinton (1966) [18]	“[…] the harmony within a man, and between a man and his world […]”	captures QoL as an ethical construct which guides modern medicine
Ware (1987) [19]	“[…] it is important to keep in mind that quality of life, as traditionally defined, is a much broader concept than health. In addition to health, quality of life encompasses standard of living, the quality of housing and the neighborhood in which one lives, job satisfaction, and many other factors. […] There is good reason to favor a more limited definition when measuring the health of an individual. The goal of the health care system is to maximize the health component of quality of life, namely health status. Measures of health outcomes should be defined accordingly.”	captures QoL as a measurement construct
World Health Organization (1997) [20]	“[…] a person’s subjective perception of their place in life in relation to the culture and value systems in which they live and in relation to their goals, expectations, standards and concerns”	captures QoL as a multifactorial concept, in which the subjective character of many factors is considered includes, besides physical and mental aspects, aspects of independence, social relationships, and personal attitude

**Table 2 jpm-12-00804-t002:** Key instrument for measuring quality of life in Parkinson’s disease.

Questionnaire	Domains
PDQ-39 ^1^ [25]	➢ Mobility ➢ Emotional well-being ➢ Stigma ➢ Social support ➢ Cognition ➢ Communication ➢ Bodily discomfort

^1^ The 39-item Parkinson’s Disease Questionnaire.

## Data Availability

Not applicable.

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
