# Peer review of "Measuring Quality of Life in Parkinson’s Disease—A Call to Rethink Conceptualizations and Assessments"

_jpm, 2022, doi:10.3390/jpm12050804_

Round 1

Reviewer 1 Report

the concept of the paper is really  important ! How and what is QoL in a personalized way is truly lacking in all chronic disease including Parkinson's. I thought as i was reading the paper that the concept of resilince would come up!  This concept addresses what seems to me to be the key point of your article , ie how does the individual sefl-adjust to the disabilities and limitations of condition like Parkinson's. 

Another point that was not clear to me is the use of the word "paradigm" .  Perhaps a more through description of the research that exists would clarify what you are trying to say is lacking in Parkinson's .

Finally, the QoL of any patient often is connected to their caregivers also not just medical care services.  This related to one of the concepts you mention "role function" which usually is related closely to "meaningfulness" .

Take a look at this ref : Santos Gracia et; Parkinsons 2021 

It would be useful to discuss right from the start why you decided to write this article . 

Author Response

Thank you for taking the time to review and comment on our manuscript.

We appreciate bringing up the important concept of resilience which we have now elaborated more detail in the manuscript. We have further provided individual domains of quality of life named by Richards et al. such as role function with examples to clarify contexts. and role function/ meaningfulness – we have addressed it a little bit more in chapter 2.2 - we also addressed the role of caregivers.

We have read the article that you have suggested with great interest. It was interesting to learn about the HY.NMSB scale. The scale is a good example, of how patients can be involved more accurately in QoL measurement – thank you for sharing!

We have also addressed the comments of other reviewers which concerned the following aspects:

  • Adding more to the objective of the manuscript
  • Grammar and spelling
  • Formulation
  • Giving concrete examples
  • Explaining the concept of “paradigm” more in detail

Reviewer 2 Report

  • Very interesting review and brings up a very important aspect of living with PD. 
  • Just a few minor edits are suggested below. 
  • Under Keywords (Line 27), I would not use the words concept or opinion. These are not usually words associated with the content in the article. Perhaps assessment, health related quality of life, etc.
  • Line 44- the word interrupt seems awkward here. Where did we go wrong (could be a possibility) or just to eliminate the question I think would also work.
  • In Table 1, in the Comments column, the first one should read "captures QoL as an ethical construct...". The second one should read, "captures QoL as a measurement construct"
  • On line 80, role function is rather vague...role of what?
  • In the last section, is it possible to include more on where we do go from here? More concrete suggestions would be more impactful. For example, on line 186, it states "taking into account additional aspects of quality of life." Such as??? On lines 192-195, I was left asking How can quantitative questionnaires more accurately depict what influences QoL of people with PD?

Author Response

Thank you very much for taking the time to review our manuscript and for your helpful comment. We have addressed your comments regarding lines 27, 44, 80, and 186 ff. accordingly.

We have also addressed the comments of other reviewers which concerned the following aspects:

  • Adding more to the objective of the manuscript
  • Grammar and spelling
  • Formulation
  • Giving concrete examples
  • Explaining the concept of “paradigm” more in detail

Reviewer 3 Report

This is a well written piece and the following are only minor observations on how the paper may be improved - mainly minor grammatical errors could be proof-read by a native english speaker

The abstract is well written and frames the piece, as is the introduction in the text. 

"Where did we interrupt" is not correct English or does not make sense to the reader - consider alternative working

Table 1 is a good addition

Section 3 is well written 

I have no major concerns

Author Response

Thank you very much for taking the time to review our manuscript and for your helpful comment. We have addressed your comment accordingly.

We have also addressed the comments of other reviewers which concerned the following aspects:

  • Adding more to the objective of the manuscript
  • Grammar and spelling
  • Formulation
  • Giving concrete examples
  • Explaining the concept of “paradigm” more in detail

Reviewer 4 Report

I appreciate the authors for presenting an important topic. However, some comments are needed to be addressed before acceptance. Here are my comments:

  • Please add a paragraph on the gap in the previous literature and how this manuscript will cover the gap. This can follow with the objectives of the current study. 
  • Please add a paragraph on the limitation of such studies and probable future usage. 

Author Response

Thank you very much for taking the time to review our manuscript and for your helpful comment.

We agree that adding a paragraph on the gap in previous literature, as well as a paragraph on the limitations of existing studies, adds quality to the manuscript and we addressed your comment in the chapters. Furthermore, we have drawn the necessary conclusions to be derived from a holistic assessment potentially applied in the future.

We have also addressed the comments of other reviewers which concerned the following aspects:

  • Grammar and spelling
  • Formulation
  • Giving concrete examples
  • Explaining the concept of “paradigm” more in detail

Round 2

Reviewer 1 Report

The paper has improved and is acceptable for publication.